# Study on the Incorporation of Chitosan Flakes in Electrospun Polycaprolactone Scaffolds

**DOI:** 10.3390/polym14081496

**Published:** 2022-04-07

**Authors:** Diana Querido, Tânia Vieira, José Luís Ferreira, Célia Henriques, João Paulo Borges, Jorge Carvalho Silva

**Affiliations:** 1Cenimat/I3N, Physics Department, School of Science and Technology, Universidade Nova de Lisboa, Caparica, 2819-516 Almada, Portugal; ds.querido@campus.fct.unl.pt (D.Q.); ts.vieira@fct.unl.pt (T.V.); jlcf@fct.unl.pt (J.L.F.); crh@fct.unl.pt (C.H.); 2Cenimat/I3N, Materials Science Department, School of Science and Technology, Universidade Nova de Lisboa, Caparica, 2819-516 Almada, Portugal

**Keywords:** hybrid scaffolds, tissue engineering, co-electrospinning, chitosan, polycaprolactone, fibroblasts, cell adhesion, cell proliferation

## Abstract

Hybrid scaffolds obtained by combining two or more biopolymers are studied in the context of tissue regeneration due to the possibility of achieving new functional properties or structural features. The aim of this work was to produce a new type of hybrid polycaprolactone (PCL)/chitosan (CS) electrospun mat through the controlled deposition of CS flakes interspaced between the PCL fibers. A poly(ethylene oxide) (PEO) solution was used to transport CS flakes with controlled size. This, and the PCL solution, were simultaneously electrospun onto a rotatory mandrel in a perpendicular setup. Different PCL/CS mass ratios were also studied. The morphology of the resulting fibers, evaluated by SEM, confirmed the presence of the CS flakes between the PCL fibers. The addition of PEO/CS fibers resulted in hydrophilic mats with lower Young’s modulus relatively to PCL mats. In vitro cell culture results indicated that the addition of CS lowers both the adhesion and the proliferation of human dermal fibroblasts. The present work demonstrates the feasibility of achieving a controlled deposition of a polymeric component in granular form onto a collector where electrospun nanofibers are being deposited, thereby producing a hybrid scaffold.

## 1. Introduction

The complexity of the extracellular matrix (ECM) of human tissues, which can be appreciated in its rich composition, diverse structures and multiple functionalities, cannot be reproduced with current scaffolding technologies [1]. Among these, electrospinning has become one of the most widely adopted due to the similarity of the scaffolds obtained using this technique with the organization of the structural proteins of the ECM [2]. In order to achieve better mimics of the ECM, blending or combining several polymers, in particular of natural and synthetic origins, is a common strategy that results in improved functionality when compared to single-component scaffolds [3,4]. The way polymers are blended or combined can also have an impact on the structural properties and biological functionality of the scaffolds. While polymer blending is the most common approach, combining different structural parts, each with its own composition, and intertwining them in a final scaffold is another approach. Valente et al. [5] report the production of electrospun nanofibers obtained from blends of chitosan (CS) and polycaprolactone (PCL) and single-component fibers of CS and of PCL that were blended in a rotating cylindrical collector. The authors show that, in some instances, scaffold properties differ even when the CS-to-PCL mass ratios are the same.

Electrospinning is a versatile technique that can be used to produce fibers of diameters ranging from tens of nanometers to micrometers. The morphology of the resulting fibers can be adjusted by changing solution and processing parameters and environmental conditions [6]. Fibers can be composed of polymers and/or ceramics, of both synthetic or natural origin, and be functionalized with a plethora of additives [7,8,9,10]. Fibrous mats have high surface area-to-volume ratio and high porosity and pore interconnectivity, being ideal substrates for cell adhesion and proliferation [11].

Chitosan (CS) is a natural, biocompatible and biodegradable polysaccharide derived from chitin that possesses mucoadhesive, anti-inflammatory, antioxidant, antimicrobial, antifungal, antihyperglycemic, antitumoral and wound-healing properties [12,13,14,15]. However, CS presents a high swelling degree and is stiff and brittle when dry while excessively soft when hydrated, limiting its application in the tissue engineering field. In order to overcome these limitations, blending CS with synthetic polymers has been analyzed [16].

Polycaprolactone (PCL) is a hydrophobic, semi-crystalline and biocompatible synthetic polyester, which degrades slowly and presents high stability, ductility and plasticity [17]. PCL can be blended with other polymers, used as a copolymer or incorporated with fillers/additives to modulate their mechanical and degradation properties [18]. Therefore, in order to synergize the individual properties of CS and PCL, blends and co-polymers have been produced and applied in skin tissue regeneration [4,19,20].

Other ways of combining CS with PCL have been studied. Nanofibrillated chitosan was loaded into PCL solutions to produce nanofibrous mats: this not only improved the mechanical properties of the structure, but also rendered the fibers hydrophilic, contributing to a higher cell viability [21]. Similar results were obtained in the study of Ji and co-workers, who added chitin nanofibrils to PCL solutions to produce electrospun fibers. The presence of nanofibrils, with a few nanometers, acted as reinforcing filler of the PCL fibers [22]. The incorporation of CS nanostructures on fibrous mats has no effect on the microscopic structure of the fibers. However, the incorporation of microscopic flakes within electrospinning solutions changed the roughness and the surface area of the final mats [23]. CS flakes with sizes between 100 and 50 µm were loaded in solutions of poly(lactic acid) (PLA) blended with poly(hydroxybutyrate) (PHB) that had been electrospun. The resulting mats fully disintegrated under composting conditions but their cytotoxicity and biocompatibility were not evaluated [24]. In the above-mentioned studies, the incorporation of CS structures in the mats was performed by loading the particles in the solution used for electrospinning. An alternative approach consists of electrospraying the chitosan solution onto the collector where nanofibers of a second polymer are being deposited. This method was used by Li et al. to assemble cellulose acetate nanofibers and chitosan particles in a layer-by-layer fashion [25]. Seddighian et al. introduced ascorbic acid within CS nanoparticles and these in a dexamethasone (Dex)-loaded PCL scaffold as a strategy for the osteogenic differentiation of mesenchymal stem cells [26]. This electrospun PCL fibrous scaffold can release Dex, as bone differentiation initiator, and ascorbic acid, as bone differentiation enhancer, in an approximately sustained release pattern for about 2 weeks. Functionalization of electrospun nanofibers with chitosan has also been studied. Cantara et al. covalently immobilized chitosan on the surface of poly(lactic-co-glycolic acid) for the culture of salivary gland epithelial cells [27]. Talukder et al. functionalized a sodium alginate/poly (vinyl alcohol) electrospun membrane with chitosan to act as an adsorbent for the removal of arsenic from water [28]. Chen et al. functionalized polyacrylonitrile nanofiber membranes with chitosan and egg white proteins for use in the treatment of dye-containing wastewater [29].

In this study, we aimed to establish whether electrospinning can be used to transport and achieve a uniform and controlled incorporation of a polymer in granular form within an electrospun scaffold. To achieve this goal, PCL fibers were obtained by electrospinning a PCL solution, and a second solution, containing polyethylene oxide (PEO), was used to transport CS flakes to ensure a consistent, homogeneous and reproducible deposition of CS on the collector. CS flakes of different sizes (0–50 µm, 50–100 µm, 100–150 µm, 150–200 µm) were dispersed in the PEO solution. The two solutions were co-electrospun at right angles onto a slowly rotating cylindrical collector. Different PCL-to-CS mass ratios were studied. To assess the possibility of using this scaffold for soft tissue engineering, the structure, mechanical properties and water contact angle were analyzed as well as the in vitro cell adhesion, proliferation and morphology.

## 2. Materials and Methods

### 2.1. Preparation of Electrospun Mats

The PCL (*M*_n_ = 70–90 kDa, from Sigma-Aldrich, St. Louis, MO, USA) solution was prepared at a concentration of 20% by mass using glacial acetic acid (from Scharlau, Barcelona, Spain) as solvent. To disperse the CS flakes (Chitopharm S, from Cognis, Monheim, Germany), a PEO (*M*_n_ = 900 kDa, from Sigma-Aldrich) solution was prepared at a concentration of 4.2% by mass in a 2:1 ethanol–distilled water mass ratio mixture.

The commercial chitosan used has flakes of sizes between 300 µm and 500 µm. To decrease and control the flake size, CS was passed through a grinder (Orbegozo MO 3300, from Sonifer, Murcia, Spain). Afterwards, the flakes obtained were separated using sieves with different opening sizes and a sieve shaker (AS 300 Control from Retsch, Haan, Germany). CS flakes with four different flake sizes were obtained: 0–50 µm, 50–100 µm, 100–150 µm and 150–200 µm. Different CS concentrations (4, 6 and 8% by mass) were dispersed in the PEO solution to achieve the PCL/CS mass ratios of 1:0.5, 1:1, 1:2 and 1:3.

The PCL solution was loaded into a 5 mL plastic syringe with a 21G stainless steel blunt needle attached. The electrospinning parameters used were: high voltage = 10 kV (Iseg T1CP300 304 p high voltage power supply, from Iseg High Voltage, Radeberg, Germany), flow rate = 0.3 mL/h (New Era NE-300 syringe pump, from New Era Pump Systems, Farmingdale, United States) and distance between needle tip and collector = 23 cm.

The PEO solution with CS flakes was loaded in a 10 mL glass syringe with an adapted pipette tip attached. The electrospinning parameters used were: high voltage = 15 kV, distance between needle tip and collector = 33 cm, and different flow rates, indicated in Table 1, were required in order to achieve the different PCL/CS mass ratios.

A cylindrical collector with a 6 cm diameter and with slow rotational and translational movements (3 rpm) was used to collect the fibers. The PCL solution was electrospun horizontally and the PEO/CS solution was electrospun vertically. Figure 1 illustrates the arrangement used while Table 1 lists the different combinations of PCL/CS mass ratios and how they were obtained by varying CS concentration in PEO solution and flow rate, and CS flake sizes studied in this work. Conversion of electrospun solution volume to polymer mass considered the density of each solution. This was measured using a 5 mL glass pycnometer, the densities obtained were: for the PCL solution, (1.229 ± 0.011) g/mL; for the solutions containing CS at 4%, (1.063 ± 0.006) g/mL; for the solution containing CS at 6%, (1.080 ± 0.006) g/mL; for the solution containing CS at 8%, (1.092 ± 0.006) g/mL. The PCL to CS mass ratio (PCL/CS) in an electrospun mat is related to the flow rate, *FR*, of the solutions, their density, *d*, and polymer concentration:$$\frac{\mathrm{PCL}}{\mathrm{CS}}\times FR_{\mathrm{CS}}\times d_{\mathrm{CS}}\times \left[\mathrm{CS}\right]=FR_{\mathrm{PCL}}\times d_{\mathrm{PCL}}\times \left[\mathrm{PCL}\right]$$

### 2.2. Characterization of Fibrous Mats

#### 2.2.1. Scanning Electron Microscopy (SEM)

A Zeiss Auriga CrossBeam electron microscope (Carl Zeiss Microscopy, Oberkochen, Germany) was used to observe the morphology of the electrospun fibers. The microscope was operated in high vacuum at 5 kV. Before observation, all samples were coated with iridium. ImageJ software (version 1.50c4, https://imagej.nih.gov/ij/, accessed on 7 February 2022) was used to measure the fiber diameter [30].

#### 2.2.2. Mechanical Characterization

Uniaxial tensile tests were performed with a tester from Rheometric Scientific with a 20 N load cell at a strain rate of 2 mm/min. Rectangular strips of 10 × 30 mm^2^ were used and their thickness was measured with a digital micrometer (Mitutoyo Corporation, Kanagawa, Japan). The Young’s modulus was calculated from the slope of the linear region of the stress–strain curve. The resulting values were determined through the average and standard deviation of at least 10 samples from three different electrospun matrices.

#### 2.2.3. Water Contact Angle (WCA)

Static water contact angle measurements were performed at room temperature and 98% humidity in a contact angle goniometer (model OCA15, DataPhysics Instruments, Filderstadt, Germany), using the sessile drop method. A 5 µL water drop was deposited on the surface of the sample and the contact angle value was acquired within the following 5 min and analyzed using the SCA15 software (version 4.3.16, DataPhysics Instruments, Filderstadt, Germany). For each sample, the contact angle was measured in at least five different locations and the results are expressed as the average ± experimental standard deviation.

### 2.3. In Vitro Evaluation of Fibrous Mats

Human fetal foreskin fibroblasts (HFFF2 cell line, obtained from ECACC, Porton Down, UK) were cultured in Dulbecco’s modified Eagle’s medium (Sigma-Aldrich), supplemented with GlutaMAX, 10% *v*/*v* foetal bovine serum (FBS), 100 units/mL of penicillin and 100 µg/mL of streptomycin, all from Life Technologies, and maintained at 37 °C in a 5% CO_2_ humidified atmosphere.

Electrospun mats with different CS flake sizes and with different PCL/CS ratios were cut out with a punch (12 mm in diameter), sterilized using ethanol 70%, washed twice with phosphate buffered saline (PBS) and soaked in complete DMEM over 24 h. The membranes were mounted in home-made Teflon inserts that were placed in 24-well plates. Cells were seeded over the fiber mats and on tissue culture polystyrene wells (TCP controls) at a density of 10 × 10^3^ cells/cm^2^. Cultures were incubated for up to 14 days and their viability was assessed.

The adhesion and proliferation of cells on the fibrous mats were measured using a resazurin (Alfa Aesar, Kandel, Germany) solution (0.2 mg/mL in PBS). Resazurin (absorption peak at 600 nm) is reduced to resorufin (absorption peak at 570 nm) by intracellular enzyme activity of viable cells. For the assay, the media were carefully removed from the samples and replaced by complete medium supplemented with 10% of resazurin solution. Seven replicas were used for each experimental condition. After 3 h of incubation in the CO_2_ incubator, the absorbance was measured at 570 nm with a reference wavelength of 600 nm using the Biotex ELX 800 V microplate reader (Biotek Instruments, Santa Clara, CA, USA). The assay was performed 24 h after cell seeding to estimate cell adhesion, and then on days 4, 7, 10, 12 and 14 of culture to assess cell proliferation.

At the end of the culture (day 14), cells growing on different substrates were stained with calcein-AM (0.4% in PBS) to observe cell distribution and morphology. All samples were examined with an epi-fluorescence microscope Nikon Eclipse Ti-S (Nikon Corporation, Tokyo, Japan).

## 3. Results

PCL and CS are usually electrospun as a blend to produce fibrous mats as substrates for cells [4,31]. Here, we designed a new method of incorporating CS flakes in PCL mats using electrospinning in order to present CS to cells in a different configuration. CS flakes, with controlled dimensions, were transported in a PEO solution and were distributed evenly in the PCL mats. Using PEO to transport CS flakes is a way to ensure that the flakes are not entrapped inside the fibers, but become exposed to cells once the PEO has been dissolved, increasing cell exposure to CS.

### 3.1. Characterization of Fibrous Mats

#### 3.1.1. Morphology of CS Flakes and Fiber Mats

Figure 2 depicts the optical microscopy images of CS flakes with different dimensions and geometries. The images confirm the different size intervals that were employed. The macroscopic structure of the fibrous mats after being electrospun were observed using a stereoscope and are represented in Figure 3, being evident the presence of CS flakes in the mats as well as the increment of the surface roughness relative to the PCL mat.

SEM images of electrospun fiber mats are shown in Figure 4. PCL fibers (Figure 4a) are smooth and free of defects with a wide fiber diameter distribution (Figure 4b). The dispersion of fiber diameters can be the result of the degradation of PCL by the acetic acid solution, which causes the hydrolysis of the ester links of PCL. As a consequence, the molar mass of the polymer is reduced, leading to a higher instability of the fiber jet and fibers with different diameters [32]. Fiber diameter ranged from 0.318 to 1.878 µm with an average of (1.16 ± 0.44) µm, similar to values reported by Ferreira et al. for PCL mats obtained from solutions of PCL dissolved in acetic acid [33].

PCL fibers with CS flakes of different sizes are shown in Figure 4c–f (decrease of flake size). CS flakes, irrespective of their sizes, were incorporated in the PCL nanofibrous scaffolds. Some flakes are visible at the surface of the mats while others situated inside the fibrous mats can be perceived through the irregular topography of the surfaces.

SEM images of PCL fibers with different CS-to-PCL mass ratios are shown in Figure 5. The CS flakes can be seen randomly distributed over the four scaffolds with different CS content. Increasing the CS ratio led to an increase in the number density of flakes over the PCL mat, as expected.

The results seen in Figure 3, Figure 4 and Figure 5 confirm the incorporation of the CS flakes in the PCL scaffolds and that electrospinning of a PEO solution containing the CS flakes is an adequate procedure for achieving an effective and even distribution of the flakes in the scaffolds.

#### 3.1.2. Mechanical Properties

Representative stress–strain curves of all nine fiber mats studied are shown in Figure 6. The Young’s modulus of each membrane is presented in Table 2.

The Young’s modulus obtained for PCL electrospun mats is (3.13 ± 0.37) MPa, while for the CS-containing membranes the values are clearly inferior. A one-way ANOVA test including the 9 types of mat yields *p* < 0.0001 and a Tukey HSD post-hoc test also results in *p* < 0.0001 for all pairs PCL vs. every other mat. The Young’s modulus of mats with CS with different flake sizes ranged from 0.79 to 0.96 MPa, increasing with the decrease of flake size. However, these variations were not significant (a one-way ANOVA test yields *p* = 0.05), indicating that changing the flake’s size has little effect on the mechanical properties of the resulting mats. The Young’s modulus decreases from 0.92 MPa to 0.24 MPa with the increment of CS content on the PCL mats. A one-way ANOVA test yields *p* < 0.001 and a Tukey HSD post-hoc test also yields *p* < 0.001 for all pairs except for mats with PCL–CS mass ratios of 0.5 vs. 1 for which *p* = 0.017 and 2 vs. 3 for which *p* = 0.00015). A higher CS content in the fiber mat structure implies a lower content per unit cross-sectional area—the PCL fibers—that can respond to the load applied, leading to greater strain for the same stress applied. This behavior is expected when particles with tens of micrometers are incorporated in fibers. Santos et al. produced poly (l-lactic acid) (PLLA) mats containing ceramic granules of irregular shape and size (≤150 µm) by dispersing the granules in the PLLA solution and showed that the presence of the granules in the PLLA nanofibers impaired the Young’s modulus but improved the stretchability of the mats when compared to the mats without granules [23]. CS microparticles (50–100 µm) loaded on PLA-PHB solutions yielded fibrous mats with a Young’s modulus lower than the mats without particles. The lower elastic modulus was attributed to the presence of beads in the fiber mats that led to lower mechanical performance [24]. The Young’s modulus of many human soft tissues lies in the interval of 0.01 MPa to 15 MPa [34,35]. In the case of skin, the slope of the stress–strain curve varies between 0.10 MPa in its initial portion (strains up to 40%) and 18.8 MPa in the high-strain region [36]. These intervals contain the Young’s moduli of the hybrid scaffolds produced, suggesting that, as far as mechanical properties are concerned, they are adequate for soft tissue engineering.

#### 3.1.3. Wettability

The mats’ wettability was determined using the water contact angle measurements. PCL fiber mats present a WCA around (132 ± 4)°, which reveals the strong hydrophobic behavior of the PCL sample prepared using acetic acid, similar to what was previously reported [5,37]. The incorporation of CS flakes in the PCL mats brings the WCA of the mats to zero. This result is probably related to the presence of water-soluble PEO fibers, which are the CS transporter, and not only with the presence of CS in the mats structure. CS is known to absorb water and its presence in the mats increases the mats’ hydrophilicity [5,38]. Therefore, both factors (the presence of CS and PEO) should contribute to the decrease of WCA values. A moderate hydrophilicity is ideal to allow the adhesion of proteins, retaining their native conformation [39].

### 3.2. In Vitro Evaluation of Mats

For the in vitro evaluation of the fibrous scaffolds, fibroblasts were used as they are ubiquitous in the human body and are the most common cell type in connective tissues.

#### Adhesion and Proliferation Assays

HFFF2 cell viability on the fibrous mats and TCP wells (cell control, CC) was assessed using the resazurin assay. The evaluations were performed 24 h after cell seeding to determine cellular adhesion and then every 3 days for up to 14 days to ascertain cell proliferation. The results are displayed in Figure 7.

The adhesion ratio (Table 3) was calculated as the quotient between the average viability of each experimental condition and the average viability of CC. Cells adhered to all membranes but at a lower ratio when compared to CC. PCL mats present an adhesion ratio of 56%, while for the mats containing CS flakes, the adhesion ranged from 32% to 45%. The differences in adhesion ratios between the PCL mat and all other mats are statistically significant (*p* < 0.0001). Fibroblast adhesion to these polymers is known to occur at low ratios [37]. The mats with different CS flake sizes exhibit adhesion ratios without significant differences between them, indicating that the size of the flakes did not affect cellular adhesion. However, the increment of CS content on the PCL mats led to the decrease of the adhesion ratios of cells, with the mats with 1:0.5 PCL/CS ratio presenting a higher adhesion rate when compared to mats with higher CS content (*p* < 0.01). The poor adhesion of cells on the mats with CS flakes can be explained by the poor adhesion properties inherent to CS itself, which are dependent on their acetylation degree and molar mass [40]. Additionally, fibrous mats with flakes are rougher structures when compared to PCL mats, impairing the adhesion and proliferation of cells as observed by Xu et al. using electrospun PLLA mats seeded with endothelial cells [41]. Furthermore, the addition of CS flakes to the mats expands the fibrous structure leading to more distant anchorage points for cells, hampering cell adhesion and leading to lower cell survival. Cell shape is a critical factor that switches cells between life and death and between proliferation and quiescence [42]. Adequate formation, distribution and maturity of focal adhesions, which mechanically bridge the scaffold to the actin cytoskeleton, are required for cell adhesion, proliferation, differentiation and further cell functions [39].

Regarding proliferation, and similar to what was stated for the adhesion rate, no significant differences are observed in cell proliferation from day 1 to day 14 on mats with different CS flake sizes (Table 3). However, the decrease in the proliferation rate with the increment of CS content in the hybrid mats is evident. This can be related to the same factors that cause a decrease in cell adhesion with increasing CS content: the inefficient adhesion of cells to CS and the spatial organization of the fibers and flakes on the mats that impact cell spreading, viability and migration. In electrospun mats of CS and silk fibroin, a lower proliferation rate of human fetal osteoblastic cells was also observed with the increment of CS content on the composite mats but higher CS content enhanced osteogenic differentiation [43].

### 3.3. Fluorescence Imaging

Figure 8 shows fluorescent images of cells stained with calcein after 14 days in culture. Cells can be seen randomly distributed on all membranes, regardless of the presence of CS flakes. Cells present a stretched morphology characteristic of fibroblasts on all mats. Cells can also be seen out of focus, below the scaffold’s surface, indicating that the expansion of the fibrous structure caused by the CS flakes created interconnected pores that allowed fibroblasts to migrate and invade the scaffold. A major concern regarding scaffolds obtained using conventional solution electrospinning is that they are unfit for tissue repair and regeneration since their small pores prevent cell infiltration [44]. The controlled deposition of a polymer in granular form may be a good method of expanding the densely packed nanofibrous structure produced by electrospinning and creating interconnected pores that allow cell infiltration and tissue ingrowth. Figure 8 also displays the autofluorescence of hybrid scaffolds that is due to the presence of CS and can be used to confirm the presence and even distribution of CS flakes along the scaffolds.

## 4. Discussion

The aim of this work was to study a new method of producing hybrid scaffolds comprising polymeric nanofibers and a polymer in granular form. PCL nanofibers were produced using conventional solution electrospinning and CS flakes were incorporated in the electrospun mat by electrospinning a PEO solution containing the CS flakes. Four different flake size ranges and four PCL to CS mass ratios were used and for all cases the hybrid scaffolds obtained revealed physical and biological that suggest these scaffolds can be useful for biomedical applications, such as wound healing. This method proved to be successful in achieving a controlled and homogeneous distribution of the flakes on the fibrous mat. It can be foreseen that this method can also be applied to ceramic particles, thereby leading to the production of composite structures. Most important is the possibility of creating an expanded fibrous structure that leads to interconnected pores with sizes allowing for cell infiltration, essential for tissue vascularization and ingrowth, which calls for further research into the application of the method described to the production of truly 3D nanofibrous scaffolds.

It is commonly accepted that scaffold degradation should ideally proceed at such a rate that newly secreted tissue replaces the scaffold as it degrades. This way, the mechanical integrity of the construct is not jeopardized, the structural support that cells need is not lost due to too-rapid scaffold degradation and the synthesis of new tissue is not inhibited by the presence of the scaffold in case the degradation is too slow. In vitro studies of polymer degradation are frequently encountered in the literature. These are normally conducted using a simulated body fluid (saline solution) or enzymatic solutions. It is very difficult to reproduce in vitro the exact conditions under which a biopolymer is degraded in vivo due to the diverse processes that lead to degradation: non-enzymatic hydrolysis, enzymatic hydrolysis and oxidative degradation. Additionally, macrophages can phagocytize small particles and break them down in lysosomes. The degree of deacetylation (DDA) also influences the biodegradation of chitosan: films and porous scaffolds produced of chitosan with DDA of 90% and higher show very low degradation (<5% mass loss) after being implanted in rats for 10–12 weeks, whereas the use of chitosan with DDA of 84% and less leads to significant mass loss (20–50%, depending on DDA) under the same conditions [45,46]. Therefore, simply by adjusting the DDA of the chitosan used, it is possible to adjust how long it takes for the chitosan flakes to be degraded in vivo, such that their presence imparts chitosan’s properties to the scaffold and the expansion of the fibrous structure that might improve cellular infiltration persists long enough for vascularization and tissue formation to occur.

## Figures and Tables

**Figure 1 polymers-14-01496-f001:**
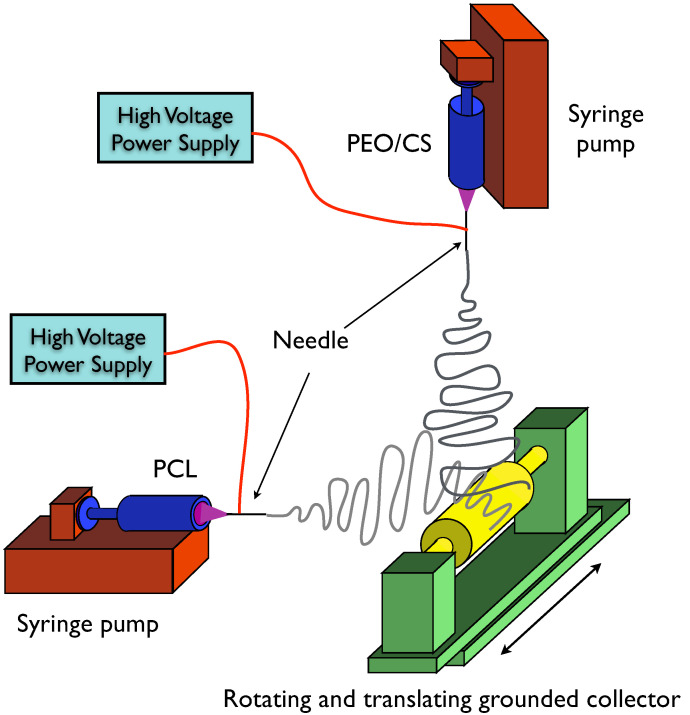
Electrospinning setup used in this work. The PCL solution was electrospun horizontally while the PEO/CS solution was electrospun vertically. The cylindrical collector made slow rotating and translating movements (3 rpm).

**Figure 2 polymers-14-01496-f002:**
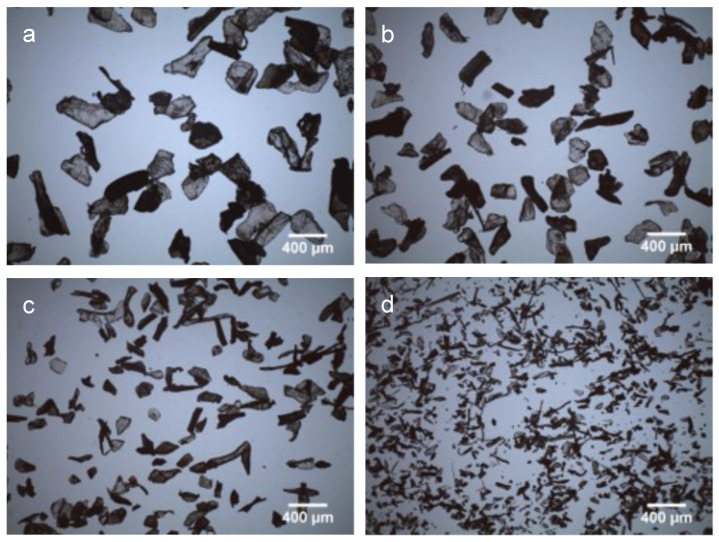
Optical microscopy images of CS flakes of different sizes: (**a**) 150–200 µm, (**b**) 100–150 µm, (**c**) 50–100 µm and (**d**) <50 µm.

**Figure 3 polymers-14-01496-f003:**
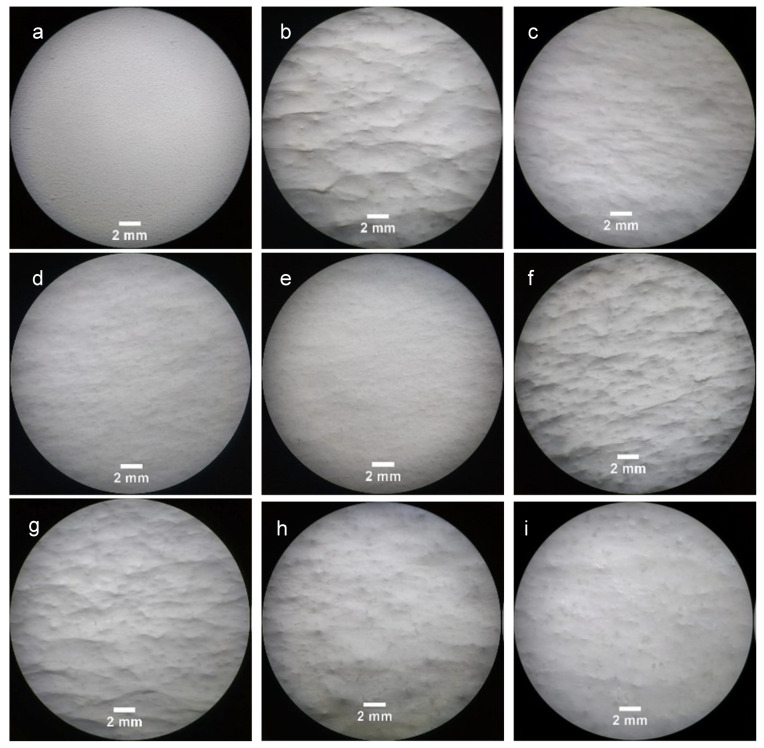
Macroscopic images of (**a**) electrospun PCL mats, PCL mats with CS flakes of different sizes: (**b**) 150–200 µm, (**c**) 100–150 µm, (**d**) 50–100 µm and (**e**) <50 µm; and PCL mats with CS flakes in the range of 50–200 µm at different mass ratios: (**f**) 1:0.5, (**g**) 1:1, (**h**) 1:2 and (**i**) 1:3.

**Figure 4 polymers-14-01496-f004:**
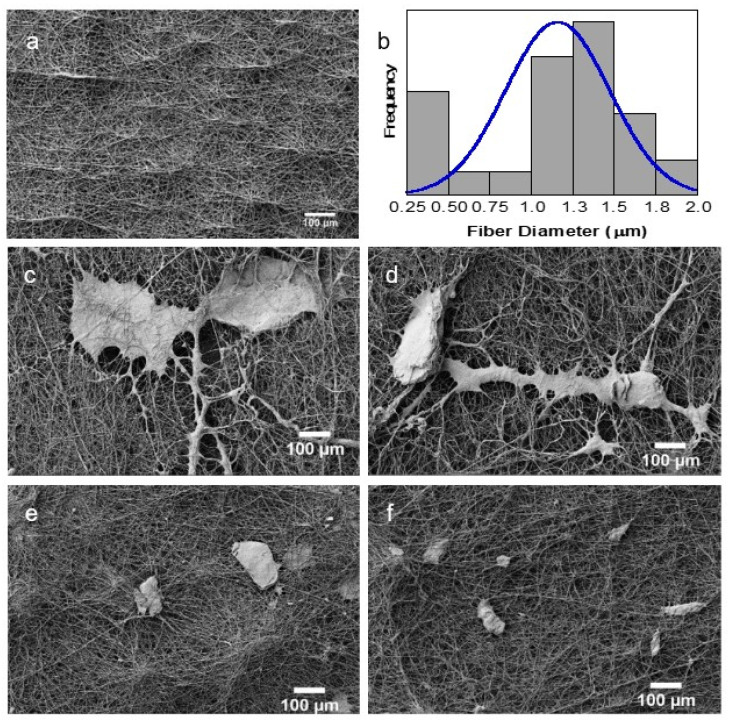
SEM images of (**a**) electrospun PCL mats and (**b**) the respective histogram with the fiber diameter distribution (**b**) as well as PCL mats with CS flakes of different sizes: (**c**) 150–200 µm, (**d**) 100–150 µm, (**e**) 50–100 µm and (**f**) <50 µm.

**Figure 5 polymers-14-01496-f005:**
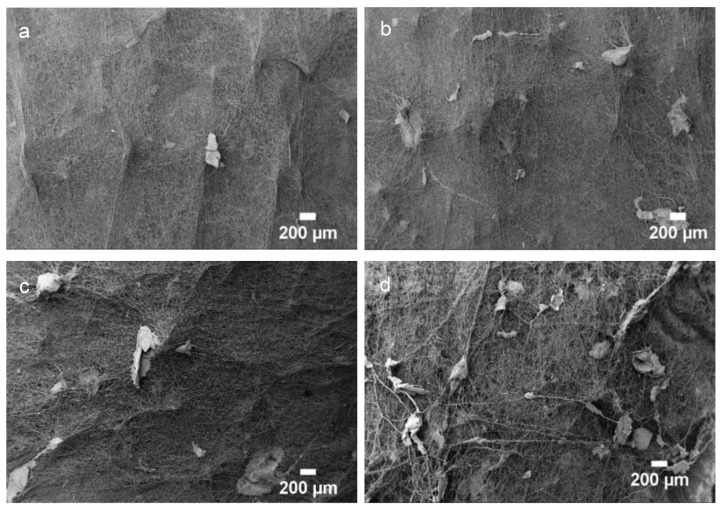
SEM images of electrospun PCL mats with CS flakes at different mass ratios: (**a**) 1:0.5, (**b**) 1:1, (**c**) 1:2 and (**d**) 1:3.

**Figure 6 polymers-14-01496-f006:**
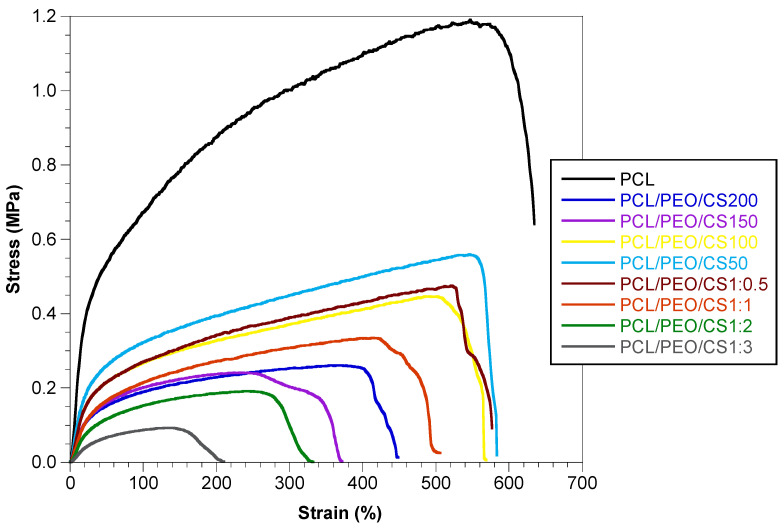
Representative stress–strain curves of PCL mats and PCL mats with CS flakes of different sizes and different PCL/CS mass ratios.

**Figure 7 polymers-14-01496-f007:**
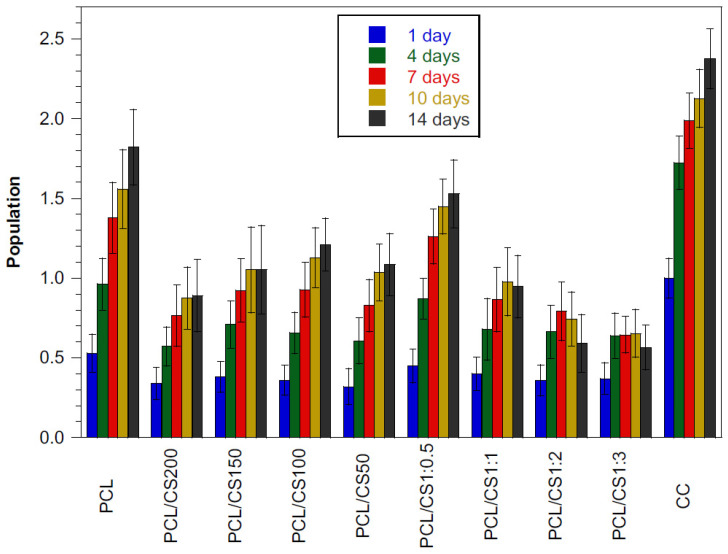
Cellular viability of HFFF2 cells seeded on PCL mats, PCL mats with CS flakes of different size ranges, PCL mats with different PCL to CS mass ratios and tissue culture plate wells (CC). Viability was evaluated using resazurin on days 1, 4, 7, 10 and 14. Vertical lines represent the standard deviation of the mean (*n* = 5).

**Figure 8 polymers-14-01496-f008:**
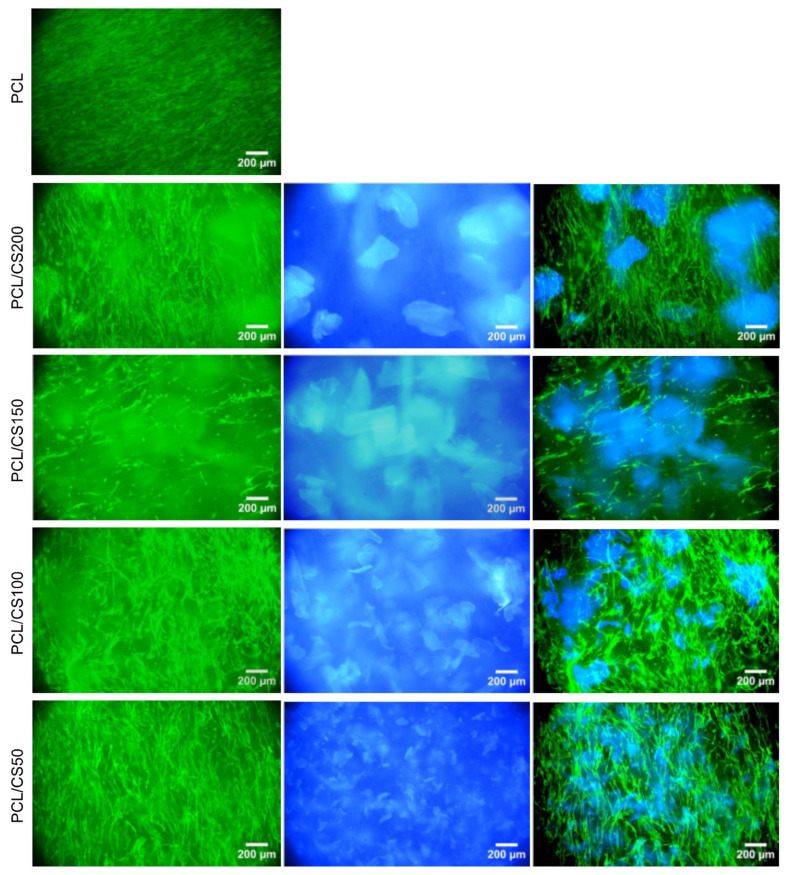
Fluorescent images of HFFF2 cells stained with calcein (green) at day 14 of culture, when growing on PCL mats, PCL mats with CS flakes of different sizes and PCL mats with CS flakes at different ratios (**left column**); autofluorescence of hybrid scaffolds when excited with UV light due to the presence of CS (**middle column**); composite images (**right column**).

**Table 1 polymers-14-01496-t001:** Mat designation, PCL-to-CS mass ratio in the electrospun mats, CS flake size, CS concentration in the PEO solutions, and flow rate of PEO/CS solution during electrospinning.

Mat Designation	PCL/CS	Flake Size/µm	[CS]/%	Flow Rate/(mL/h)
PCL/PEO/CS200	1:1.15	150–200	4	2
PCL/PEO/CS150	1:1.15	100–150	4	2
PCL/PEO/CS100	1:1.15	50–100	4	2
PCL/PEO/CS50	1:1.15	0–50	4	2
PCL/PEO/CS1:0.5	1:0.5	50–200	4	0.87
PCL/PEO/CS1:1	1:1	50–200	4	1.73
PCL/PEO/CS1:2	1:2	50–200	6	2.28
PCL/PEO/CS1:3	1:3	50–200	8	2.52

**Table 2 polymers-14-01496-t002:** Values of mechanical properties (Young’s modulus, *Y*) and water contact angle (WCA) of PCL electrospun mats with and without CS flakes.

Mat	*Y*/MPa	WCA/°
PCL	3.13 ± 0.37	132 ± 4
PCL/PEO/CS200	0.79 ± 0.16	0
PCL/PEO/CS150	0.84 ± 0.12	0
PCL/PEO/CS100	0.94 ± 0.12	0
PCL/PEO/CS50	0.96 ± 0.18	0
PCL/PEO/CS1:0.5	0.92 ± 0.17	0
PCL/PEO/CS1:1	0.78 ± 0.08	0
PCL/PEO/CS1:2	0.45 ± 0.06	0
PCL/PEO/CS1:3	0.24 ± 0.04	0

**Table 3 polymers-14-01496-t003:** Adhesion ratio of HFFF2 cells to the electrospun PCL mats with CS flakes from different sizes and in different ratios and cell proliferation obtained by the ratio between cell population on day 14 and on day 1 (PR14/1). Uncertainty is the combined standard uncertainty.

Mat Designation	Adhesion Ratio/%	PR14/1
PCL	56 ± 11	3.4 ± 0.3
PCL/CS200	35 ± 9	2.7 ± 0.3
PCL/CS150	39 ± 9	2.8 ± 0.3
PCL/CS100	38 ± 9	3.3 ± 0.2
PCL/CS50	36 ± 10	3.3 ± 0.2
PCL/CS1:0.5	47 ± 9	3.4 ± 0.2
PCL/CS1:1	42 ± 10	2.4 ± 0.2
PCL/CS1:2	38 ± 9	1.6 ± 0.2
PCL/CS1:3	37 ± 9	1.5 ± 0.2
CC	100 ± 8	2.4 ± 0.2

## Data Availability

Data are contained within the article. Raw data generated during this work are available from the corresponding author upon reasonable request.

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
