# Peer review of "Study on the Incorporation of Chitosan Flakes in Electrospun Polycaprolactone Scaffolds"

_polymers, 2022, doi:10.3390/polym14081496_

Round 1

Reviewer 1 Report

the study presented by Diana Querido et al, entitled: Study on the incorporation of chitosan flakes in electrospun 2 polycaprolactone scaffolds, is an interesting proposal for the development of a biomaterial that we could consider mixed since it starts from a PCL scaffold in which they deposit CS flakes. Both biomaterials have already demonstrated their usefulness in the world of tissue engineering for several years, with good and sometimes not so good results (fibrosis formation).
There are some points of interest that should be named:

  1. The physical characteristics such as the mechanical properties or the size of the flake, for what purposes make it a candidate, as a support for the regeneration of dermal, bone, and nervous tissue? an important point in these characteristics is that they have the same mechanical characteristics and memory of the tissues where they will be implanted, does it meet this requirement?
  2. because use fibroblasts and not a line of vascular tissue, one of the points of interest in tissue regeneration is that the biomateial allows vascular growth since it is essential for the metabolic support of the tissue in the process of regeneration
  3. Does the author know how long it takes for the CS to degrade, does it last the necessary time for the purpose that it proposes to use it?
  4. The author with the data he has presents a very brief discussion, he should take advantage of the data and address the points mentioned above in the discussion

Thanks

Author Response

The authors thank the reviewer for carefully reading our manuscript and for the constructive criticism provided.

Our answers to the reviewer’s queries follow.

Reviewer 1

  1. The physical characteristics such as the mechanical properties or the size of the flake, for what purposes make it a candidate, as a support for the regeneration of dermal, bone, and nervous tissue? an important point in these characteristics is that they have the same mechanical characteristics and memory of the tissues where they will be implanted, does it meet this requirement?

The mechanical properties of the hybrid scaffolds lie in the interval reported in the literature as typical of human soft tissues. We added the following paragraph (lines 269-274) to the manuscript to discuss this property:

“ The Young’s muduli of most human soft tissues lie in the interval 0.01 MPa to 15 MPa [34,35]. In the case of skin, the slope of the stress–strain curve varies between 0.10 MPa in its initial portion (strains up to 40%) and 18.8 MPa in the high strain region [36]. These intervals contain the Young’s moduli of the hybrid scaffolds produced, suggesting that, as far as mechanical properties are concerned, they are adequate for soft tissue engineering.”

  1. because use fibroblasts and not a line of vascular tissue, one of the points of interest in tissue regeneration is that the biomateial allows vascular growth since it is essential for the metabolic support of the tissue in the process of regeneration

Vascularization is indeed a central issue in Tissue Engineering. Given its importance, the authors made a final comment in the discussion section that future work should also evaluate the ability of endothelial cells to penetrate the scaffold and vascularize the new tissue being regenerated. The focus of this work was the incorporation of a second polymer in granular form within the fibrous scaffold made of another polymer. Therefore, for the in vitro tests, fibroblasts were used as they are ubiquitous in the human body and the most common cell type in connective tissues. The justification for the use of fibroblasts was included in the manuscript (lines 294-295) 

  1. Does the author know how long it takes for the CS to degrade, does it last the necessary time for the purpose that it proposes to use it?

Scaffold degradation should ideally proceed at such a rate that newly secreted tissue replaces the scaffold as it degrades so that mechanical integrity of the construct is not jeopardized, the structural support that cells need is not lost due to too rapid scaffold degradation and the synthesis of new tissue is not inhibited by the presence of the scaffold if degradation is too slow. In vitro studies of polymer degradation are frequently encountered in the literature. These are normally made using a simulated body fluid (saline solution) or enzymatic solutions. It is very difficult to reproduce in vitro the exact conditions under which a biopolymer is degraded in vivo due to the diverse processes that lead to degradation: non-enzymatic hydrolysis, enzymatic hydrolysis and oxidative degradation. Also, macrophages can phagocytize small particles and break them down in lysosomes. The degree of deacetylation (DDA) also influences the biodegradation of chitosan: films and porous scaffolds produced of chitosan with DDA of 90% and higher, show very low degradation (<5% mass loss) after being implanted in rats for 10–12 weeks, whereas the use of chitosan with DDA of 84% and less leads to significant mass loss (20–50%, depending on DDA) under the same conditions [1,2]. Therefore, simply by adjusting the DDA of the chitosan used, it is possible to adjust how long it takes for the chitosan flakes to be degraded in vivo such that their presence impart to the scaffold chitosan’s properties (mucoadhesive, anti-inflammatory, antioxidant, antimicrobial, antifungal, antihyperglycemic, antitumoral and wound healing enhancer [3]) and the expansion of the fibrous structure that might improve cellular infiltration persists long enough for vascularization and tissue formation to occur.

  1. The author with the data he has presents a very brief discussion, he should take advantage of the data and address the points mentioned above in the discussion

The discussion section is short due to the fact that results were discussed along their presentation. Following the reviewer’s suggestion, we included the comments above regarding biodegradation of chitosan in the discussion section.

[1]  K. Tomihata, Y. Ikada, In vitro and in vivo degradation of films of chitin and its deacetylated derivatives, Biomaterials. 18 (1997) 567–575. doi:10.1016/S0142-9612(96)00167-6.

[2] Y. Wan, A. Yu, H. Wu, Z. Wang, D. Wen, Porous-conductive chitosan scaffolds for tissue engineering II. In vitro and in vivo degradation, J. Mater. Sci. Mater. Med. 16 (2005) 1017–1028. doi:10.1007/s10856-005-4756-x.

[3] I. Aranaz, A.R. Alcántara, M.C. Civera, C. Arias, B. Elorza, A.H. Caballero, N. Acosta, Chitosan: An overview of its properties and applications, Polymers (Basel). 13 (2021). doi:10.3390/polym13193256.

Reviewer 2 Report

The work “Study on the incorporation of chitosan flakes in electrospun polycaprolactone scaffolds” presents a representative analysis of the proposed study and I recommend publication after the following corrections:

INTRODUCTION

#Line 37:

Please check the citation format. Is [3][4] or [3,4]? Check the entire document.

#Line 41:

The sentence “Valente et.al. electrospun blends of chitosan (CS) and polycaprolactone (PCL)…” is incomplete. Please, reformulated, for example: Valente et.al. [5] “reports”, “demonstrates”, “study” an electrospun…

MATERIALS AND METHODS

#Line 95:

Please insert the numbering in the subsection. Check the entire document.

#Line 106 to 107:

Join this sentence to the previous paragraph.

#Line 111:

How the operational parameters were defined?

#Table 1:

How were the mass proportions defined? Also, does the "50-200" flake size mean it was tested on the same ranges as before or not?

RESULTS AND DISCUSSION

The authors mention the possible application of the material obtained as a wound dressing (biomedical area). In this sense, would not it be interesting to perform solubility analysis or swelling index?

#Line 239:

Why not evaluate tensile strength and elongation percentage at break?

#Line 245:

There is a statistical analysis on the results that supported this claim.

#Line 362:

This topic should not be called "conclusion"?

Author Response

The authors thank the reviewer for carefully reading our manuscript and for the constructive criticism provided.

Reviewer 2

#Line 37:

Please check the citation format. Is [3][4] or [3,4]? Check the entire document.

Done

#Line 41:

The sentence “Valente et.al. electrospun blends of chitosan (CS) and polycaprolactone (PCL)…” is incomplete. Please, reformulated, for example: Valente et.al. [5] “reports”, “demonstrates”, “study” an electrospun…

The sentence has been reworded and split into two sentences to make it clearer and more readable: “. Valente et.al. [1]report the production of electrospun nanofibers obtained from blends of chitosan (CS) and polycaprolactone (PCL) and single-component fibers of CS and of PCL that were blended in a rotating cylindrical collector. The authors show that, in some instances, scaffold properties differ even when the CS to PCL mass ratios are the same.”

MATERIALS AND METHODS

#Line 95:

Please insert the numbering in the subsection. Check the entire document.

Done

#Line 106 to 107:

Join this sentence to the previous paragraph.

Done

#Line 111:

How the operational parameters were defined?

The operational parameters were chosen based on our previous work involving PCL dissolved in glacial acetic acid [1][2][3] and PEO dissolved in ethanol:water mixtures [4].

#Table 1:

How were the mass proportions defined?

The mass proportions of the first 4 mats listed in table 1 were calculated from the PCL and CS concentrations, solution densities and the flow rates employed. As an example, for the first mat, since PCL solutions have polymer concentration of 20% and were spun at a rate of 0.3 mL/h and their density was measured at 1.229 g/mL, PCL is deposited at a rate of 0.3 ml/h x 1.229 g/ml x 0.2 = 0.0737 g/h. The PEO/CS solution has a density of 1.063 g/ml and CS concentration is 4%: therefore, CS is deposited at a rate of 2 mL/h x 1.063 g/mL x  0.04 = 0.085. Finally, the PCL to CS mass ratio is 0.0737:0.085 = 1:1.15

This calculation requires knowledge of solution density, which we measured, but did not report in the original version of the manuscript. So that any reader may repeat our calculation, we included the following sentence starting line 133: “This was measured using a 5 mL glass pycnometer, the densities obtained were: for the PCL solution, 1.229±0.011 g/mL; for the solutions containing CS at 4%, 1.063±0.006 g/mL; for the solution containing CS at 6%, 1.080±0.006 g/mL; for the solution containing CS at 8%, 1.092±0.006 g/mL. The PCL to CS mass ratio (PCL/CS) in an electrospun mat is related to the flow rate, FR, of the solutions, their density, d, and polymer concentration:

Also, does the "50-200" flake size mean it was tested on the same ranges as before or not?

This means that in the case of the final 4 mats listed in table 1 the CS flake size used was in the interval between 50 µm and 200 µm. To help clarify this meaning we added the following to the caption of figure 2: “in the range 50 – 200 µm”

RESULTS AND DISCUSSION

The authors mention the possible application of the material obtained as a wound dressing (biomedical area). In this sense, would not it be interesting to perform solubility analysis or swelling index?

The reviewer mentions two important properties of a wound dressing, solubility and swelling, that weren’t measured in the current study, but to which the authors will give due attention in future studies.

The solubility (degradability) of PCL is known to occur at a very slow rate

due to PCL’s hydrophobic nature and the absence of enzymes capable of efficiently degrading it in vivo. Degradation of bulk PCL in vivo lasts years and begins with chain scission by hydrolytic cleavage of ester groups [5]. When the particle size of the implant decreases due to fragmentation or when powdered samples are implanted, PCL is rapidly degraded in a few days inside the phagosomes of macrophages and giant cells [6].

Chitosan degradation can occur through different mechanisms such as acid hydrolysis, oxidative–reductive or enzymatic degradation using specific and non-specific enzymes. In vitro degradation rates reported depend strongly on the type and concentration of the agent studied. In vivo degradation studies are more difficult to perform. Chitosan with a degree of acetylation of 50% degraded completely in 2 weeks, with degradation rate decreasing with decreasing acetylation degree [7]. Within the 30 days of implantation in rabbit’s femurs, chitosan freeze-dried scaffolds showed complete biodegradation [8].

Regarding the swelling index, again, because PCL is a hydrophobic polymer no swelling is expected that is attributable to PCL. CS only dissolves in acidic solutions but swells in neutral or slightly acidic solutions. Swelling indexes of a few hundred percent have been reported: these will cause some swelling of the hybrid scaffolds produced in this study.

A discussion on the topic of scaffold degradation was added to the discussion section.

#Line 239:

Why not evaluate tensile strength and elongation percentage at break?

This information was not extracted from the stress-strain curves obtained because all hybrid scaffolds have an elongation at break above 100% and we believe that in practical terms this guarantees that the membranes will endure significant deformation without rupture during a normal use.

#Line 245:

There is a statistical analysis on the results that supported this claim.

Yes, it was added to the discussion on mechanical properties. A one-way ANOVA of the mechanical properties of those 4 scaffolds results in a p = 0.05. Given that the p value lies at the border of the commonly used threshold between statistically significant and non-significant results, and a threshold of 0.005 has been proposed [9], we opted to say the differences are not significant. We also added the statistical analysis results for the other comparisons made.

#Line 362:

This topic should not be called "conclusion"?

We followed the MDPI template for the journal Polymers. The conclusions section “is not mandatory but can be added to the manuscript if the discussion is unusually long or complex.” Since our discussion section is relatively short as we discussed the results along with their presentation, we did not include a “Conclusion” section.

[1]      T. Valente, J.L. Ferreira, C. Henriques, J.P. Borges, J.C. Silva, Polymer blending or fiber blending: A comparative study using chitosan and poly(ε-caprolactone) electrospun fibers, J. Appl. Polym. Sci. 136 (2019) 1–11. doi:10.1002/app.47191.

[2]      S.R.R. Gomes, G. Rodrigues, G.G. Martins, M.A. Roberto, M. Mafra, C.M.R. Henriques, J.C. Silva, In vitro and in vivo evaluation of electrospun nanofibers of PCL, chitosan and gelatin: A comparative study, Mater. Sci. Eng. C. 46 (2015) 348–358. doi:10.1016/j.msec.2014.10.051.

[3]      S. Gomes, D. Querido, J.L. Ferreira, J.P. Borges, C. Henriques, J.C. Silva, Using water to control electrospun Polycaprolactone fibre morphology for soft tissue engineering, J. Polym. Res. 26 (2019) 222. doi:10.1007/s10965-019-1890-6.

[4]      C. Henriques, R. Vidinha, D. Botequim, J.P. Borges, J.A.M.C. Silva, A Systematic Study of Solution and Processing Parameters on Nanofiber Morphology Using a New Electrospinning Apparatus, J. Nanosci. Nanotechnol. 9 (2009) 3535–3545. doi:10.1166/jnn.2009.NS27.

[5]      C.G. Pitt, F.I. Chasalow, Y.M. Hibionada, D.M. Klimas, T. Park, N. Carolina, Aliphatic Polyesters . I . The Degradation of Poly ( e- caprolactone ) In Vivo, J. Appl. Polym. Sci. 26 (1981) 3779–3787. doi:10.1002/app.1981.070261124.

[6]      S.C. Woodward, P.S. Brewer, F. Moatamed, A. Schindler, C.G. Pitt, The intracellular degradation of poly(epsilon-caprolactone), J. Biomed. Mater. Res. 19 (1985) 437–444. doi:10.1002/jbm.820190408.

[7]      S.M. Lim, D.K. Song, S.H. Oh, D.S. Lee-Yoon, E.H. Bae, J.H. Lee, In vitro and in vivo degradation behavior of acetylated chitosan porous beads, J. Biomater. Sci. Polym. Ed. 19 (2008) 453–466. doi:10.1163/156856208783719482.

[8]      S.B. Qasim, S. Husain, Y. Huang, M. Pogorielov, V. Deineka, M. Lyndin, A. Rawlinson, I.U. Rehman, In-vitro and in-vivo degradation studies of freeze gelated porous chitosan composite scaffolds for tissue engineering applications, Polym. Degrad. Stab. 136 (2017) 31–38. doi:10.1016/j.polymdegradstab.2016.11.018.

[9]      D.J. Benjamin, J.O. Berger, M. Johannesson, B.A. Nosek, E.J. Wagenmakers, R. Berk, K.A. Bollen, B. Brembs, L. Brown, C. Camerer, D. Cesarini, C.D. Chambers, M. Clyde, T.D. Cook, P. De Boeck, Z. Dienes, A. Dreber, K. Easwaran, C. Efferson, E. Fehr, F. Fidler, A.P. Field, M. Forster, E.I. George, R. Gonzalez, S. Goodman, E. Green, D.P. Green, A.G. Greenwald, J.D. Hadfield, L. V. Hedges, L. Held, T. Hua Ho, H. Hoijtink, D.J. Hruschka, K. Imai, G. Imbens, J.P.A. Ioannidis, M. Jeon, J.H. Jones, M. Kirchler, D. Laibson, J. List, R. Little, A. Lupia, E. Machery, S.E. Maxwell, M. McCarthy, D.A. Moore, S.L. Morgan, M. Munafó, S. Nakagawa, B. Nyhan, T.H. Parker, L. Pericchi, M. Perugini, J. Rouder, J. Rousseau, V. Savalei, F.D. Schönbrodt, T. Sellke, B. Sinclair, D. Tingley, T. Van Zandt, S. Vazire, D.J. Watts, C. Winship, R.L. Wolpert, Y. Xie, C. Young, J. Zinman, V.E. Johnson, Redefine statistical significance, Nat. Hum. Behav. 2 (2018) 6–10. doi:10.1038/s41562-017-0189-z.

Reviewer 3 Report

The authors present a method to incorporate chitosan flakes of different sizes in the electrospun mat to produce wound dressing. Different rations of CS to PCL were analyzed. The CS microparticles were transported with polyethylene oxide in a co-electrospinning setup.

In general, this method shows that many problems need to be resolved before finalizing the preparation of wound dressing.

Specific comments:

  1. Is the chitosan particle or chitosan solution electrospinning the only way to produce electrospun mats containing chitosan? There is also chitosan functionalization method that could be used.
  2. Since the Authors study the impact of CS microparticle size on the electrospinning, it would be preferential to format the introduction section in a way that shows that, e.g. Li et al. with CS microparticles (of what micro size) example and then with decreasing size of the particle, finally leading to an example with e.g. Ji et al. that electrospun nanofibrillated CS (of what nanosize). Now it is somewhat mixed, with first mentioning nano, then micro and finally also nanoparticles of chitosan. There should be a kind of order.
  3. Why did the Authors use glacial acetic acid for PCL solubilization? What is the rationale for it? It just caused many problems with chain degradation.
  4. Based on my calculations, the PCL/CS for the first four materials in Table 1 should be 1:33 not 1:15 s. Can the Authors show how they calculated this ratio?
  5. Line 168. How does the sterilization with ethanol and PBS soaking affect PEO removal? Is it possible that PEO is still trapped in the material and affecting cells?
  6. Another interesting method for CS flakes incorporation that does not include adding probably cytotoxic PEO, is combining electrospinning of PCL and electrospraying of CS flakes. This will give similar results of material roughness etc.
  7. Line 185. Did the Authors use DAPI?
  8. Figure 3 and 2 have different (a) (b) ... formatting. One letter is after the part describing figure (Fig 2), once it is before like in Fig 3.
  9. Figure 3, are those spots on Fig 3 c and d flakes or simply remainings of PEO?
  10. Line 234 How did the Authors measure the even distribution of flakes in mat? How would they define it?
  11. Line 289. This means that having no material will give an ideal WVTR.
  12. Line 315 I don't agree that flakes incorporation make it more difficult for cell anchorage. The material gets rougher, but the scale of it is different. Cells are just 20-30 micrometres.
  13. Table 3, Cell control is depicted as 100%, but the truth is that also for TCP control, not all cells will attach, and usually, it gives 90-95% of adhesion degree.
  14. Images in Figure 7 should be replaced with ones done with the confocal microscope. It just shows autofluorescence from the fibres and the flakes.
  15. Line 366-368 The authors did not perform any in vivo study with wounds; therefore, such statement should be rewritten.
  16. Line 372 The authors also prepared a composite. Please rewrite it.
  17. Line 373 Please perform the confocal study with actin/nuclei staining to confirm such a statement.

Author Response

The authors thank the reviewer for carefully reading our manuscript and for the constructive criticism provided.

Our answers to the reviewer’s queries follow.

  1. Is the chitosan particle or chitosan solution electrospinning the only way to produce electrospun mats containing chitosan? There is also chitosan functionalization method that could be used.

In fact, functionalization of an electrospun membrane with chitosan is another approach to the incorporation of chitosan. Therefore, we added the following sentences to the introduction: “Functionalization of electrospun nanofibers with chitosan has also been studied. Cantara et.al. covalently immobilized chitosan on the surface of poly(lactic-co-glycolic acid) for the culture of salivary gland epithelial cells [27]. Talukder et.al. functionalized a sodium alginate/poly(vinyl alcohol) electrospun membrane with chitosan to act as an adsorbent for the removal of arsenic from water [28]. Chen et.al. functionalized polyacrylonitrile nanofiber membranes with chitosan and egg white proteins for use in the treatment of dye-containing wastewater [29].”

  1. Since the Authors study the impact of CS microparticle size on the electrospinning, it would be preferential to format the introduction section in a way that shows that, e.g. Li et al. with CS microparticles (of what micro size) example and then with decreasing size of the particle, finally leading to an example with e.g. Ji et al. that electrospun nanofibrillated CS (of what nanosize). Now it is somewhat mixed, with first mentioning nano, then micro and finally also nanoparticles of chitosan. There should be a kind of order.

The sequence in which the publications mentioned by the reviewer are presented in the introduction follow a logical order, going from the incorporation of CS particles (from nano to micro) in the electrospinning solutions to the electrospraying of CS (nano)particles. First, we mention the fabrication of electrospun nanofibers containing nanofibrillated CS (Fadaie et.al.) and CS nanofibrils (Ji et.al.). Then, we present the work of Santos et.al. who produced PLLA mats containing ceramic microgranules by dispersing them in the PLLA solution and finally the work of Arrieta et.al. who loaded CS microparticles on PLA-PHB solutions.

Finally, we present the work of Li et.al., where the size of the chitosan particles is not determined but appears to be mostly nanometer sized, and then the work of Seddighian et.al. who encapsulated ascorbic acid within CS nanoparticles .

  1. Why did the Authors use glacial acetic acid for PCL solubilization? What is the rationale for it? It just caused many problems with chain degradation.

We used acetic acid as a solvent for PCL because in our previous studies on PCL electrospinning we found it easier to produce fibers with smaller diameter [1].

  1. Based on my calculations, the PCL/CS for the first four materials in Table 1 should be 1:33 not 1:15 s. Can the Authors show how they calculated this ratio?

To answer this question, we repeat here the answer to reviewer 2, who raised an identical question.

The mass proportions of the first 4 mats listed in table 1 were calculated from the PCL and CS concentrations, solution densities and the flow rates employed. As an example, for the first mat, since PCL solutions have polymer concentration of 20% and were spun at a rate of 0.3 mL/h and their density was measured at 1.229 g/mL, PCL is deposited at a rate of 0.3 ml/h x 1.229 g/ml x 0.2 = 0.0737 g/h. The PEO/CS solution has a density of 1.063 g/ml and CS concentration is 4%: therefore, CS is deposited at a rate of 2 mL/h x 1.063 g/mL x  0.04 = 0.085. Finally, the PCL to CS mass ratio is 0.0737:0.085 = 1:1.15

This calculation requires knowledge of solution density, which we measured, but did not report in the original version of the manuscript. So that any reader may repeat our calculation, we included the following sentence starting line 133: “This was measured using a 5 mL glass pycnometer, the densities obtained were: for the PCL solution, 1.229±0.011 g/mL; for the solutions containing CS at 4%, 1.063±0.006 g/mL; for the solution containing CS at 6%, 1.080±0.006 g/mL; for the solution containing CS at 8%, 1.092±0.006 g/mL. The PCL to CS mass ratio (PCL/CS) in an electrospun mat is related to the flow rate, FR, of the solutions, their density, d, and polymer concentration:

  1. Line 168. How does the sterilization with ethanol and PBS soaking affect PEO removal? Is it possible that PEO is still trapped in the material and affecting cells?

PEO is soluble in both ethanol and water; therefore, it is reasonable to assume that after soaking the membranes in ethanol, twice in PBS and then in culture medium no PEO remains inside the scaffolds and cannot have any negative impact on cells. Moreover, PEO is a biocompatible polymer.

  1. Another interesting method for CS flakes incorporation that does not include adding probably cytotoxic PEO, is combining electrospinning of PCL and electrospraying of CS flakes. This will give similar results of material roughness etc.

It is, in fact, an interesting and alternative approach to the incorporation of chitosan microparticles in the scaffold. This method is mentioned in the introduction, lines 79-86. We also studied the production of CS microparticles by electrospraying a 3% CS solution into a 1 M NaOH coagulation bath and obtained particles with an average diameter between 93 µm and 643 µm when using voltages between 19.5 kV and 6 kV (unpublished).

  1. Line 185. Did the Authors use DAPI?

No, DAPI cannot be used because DAPI is excited in the UV range and chitosan fluoresces strongly when illuminated with UV light.

  1. Figure 3 and 2 have different (a) (b) ... formatting. One letter is after the part describing figure (Fig 2), once it is before like in Fig 3.

The caption of figure 2 was changed to follow the format of figures 1, 3 and 4.

  1. Figure 3, are those spots on Fig 3 c and d flakes or simply remainings of PEO?

The spots are chitosan flakes still embedded in PEO and some traces of PEO.

  1. Line 234 How did the Authors measure the even distribution of flakes in mat? How would they define it?

We did not try to quantify the distribution of flakes in the mat. Our conclusion is based on the observation of the distribution of CS flakes that can be seen in both optical (including fluorescence) and SEM images of the surfaces.

  1. Line 289. This means that having no material will give an ideal WVTR.

The pertinent comment by the reviewer led us to re-analyze our results for the WVTR. Given that the calculations were all checked and no errors were found, we turned to the experimental procedure. A repetition of the reference (open glass vial) kept in the desiccator and oven over a period of 15 hours gave a WVTR of 7130 g/m2/day. When left at ambient temperature, the WVTR is close to the one reported in the original version of the manuscript. This tells us that our procedure did not allow for an adequate establishment of a steady state at a temperature of 37°C. Therefore, the results we report were obtained at an (unknown) equivalent temperature that was closer to ambient temperature than to 37°C. This, and the fact that a determination of the WVTR of a wound dressing is meaningful only if made at 37°C, we decided to remove the WVTR results from our manuscript.

  1. Line 315 I don't agree that flakes incorporation make it more difficult for cell anchorage. The material gets rougher, but the scale of it is different. Cells are just 20-30 micrometres.

Our results indicate a statistically significant different in cell adhesion ratio between PCL scaffolds and all other scaffolds containing CS flakes.

The authors try to explain this fact using results reported in the literature:

  • cell adhesion to nanofibrous chitosan scaffolds is lower than cell adhesion to PCL nanofibrous scaffolds [2];
  • rougher structures (electrospun fibers vs. solvent cast films) lead to decreased endothelial cell adhesion and proliferation [3].

While the surface roughness of the scaffolds is, as the reviewer points, of a very different scale from that of cells, this roughness arises from the presence of CS flakes that cause a separation of the fibers (that cannot, unfortunately, be perceived in SEM images due to the very nature of the technique) and, as we discuss, the anchorage points available for cells become more distant and this reduces the chances of cells establishing the required focal adhesions necessary for cell survival.

To help clarify and support our reasoning, we moved lines 336-338 that were in the paragraph discussing cell proliferation are now lines 314-316 in the revised version.

  1. Table 3, Cell control is depicted as 100%, but the truth is that also for TCP control, not all cells will attach, and usually, it gives 90-95% of adhesion degree.

While it is true that not all cells adhere to TCP, these are mainly cells that suffered some damage during processing (from trypsinization to seeding) and these are also present in the population of cells that are seeded on the scaffolds. Therefore, it is reasonable to assume that all viable cells will adhere to TCP and calling this a 100% adhesion ratio is not only acceptable but also common practice.

  1. Images in Figure 7 should be replaced with ones done with the confocal microscope. It just shows autofluorescence from the fibres and the flakes.

PCL has no autofluorescence and CS has autofluorescence mainly in the UV region (also some in the green and orange channels). This property enables us to locate the CS flakes in the scaffold, providing a further indication of the homogeneous distribution of CS in the scaffolds (middle and left columns in figure 7). The green structures in figure 7 are the viable cells stained with calcein. Some background comes from light dispersion and CS autofluorescence. Given the aim of this test - prove the presence and viability of cells on these scaffolds - we think these images prove the point.

  1. Line 366-368 The authors did not perform any in vivo study with wounds; therefore, such statement should be rewritten.

As the reviewer notices, the authors did not perform in vivo studies. Therefore, this statement is written in hypothetical terms: our results suggest that biomedical applications, like wound healing, might be feasible with these hybrid scaffolds. In our opinion, no definite statements about this type of application are made and the sentence is not misleading.

  1. Line 372 The authors also prepared a composite. Please rewrite it.

The authors use the term composite as denoting “a material made from two or more constituent materials with significantly different physical or chemical properties which remain separate and distinct at the macroscopic or microscopic scale within the finished structure”. In this sense, the authors did not produce a composite material, as PCL and CS do not have significantly different properties but produced a hybrid material instead.

  1. Line 373 Please perform the confocal study with actin/nuclei staining to confirm such a statement.

This statement is made in the context of a discussion of future work where the properties and features of these scaffolds should be further evaluated. We raise the hypothesis that the incorporation of CS in granular form will expand the fibrous structure to the point that cells can effectively invade the whole scaffold’s volume and vascularization of the scaffold may occur for this is paramount to tissue regeneration. Further research should investigate this issue and, if proven in vitro, perform in vivo tests in wounds to evaluate the performance of the scaffold in assisting skin regeneration.

Reviewer 4 Report

The current manuscript provides a confusing account of chitosan flakes + PCL nanofibers. There are several concerns with the study as follows:

1. The method is not clear at all. The PCl solution and the CS/PEO solution were separately prepared and separately electrospun. Why then mention the 5ml/1-ml syringe? Is it a layered system? or a biaxial system etc.

2. The rationale and motivation of the study is not clear. Why are the authors using flakes... what are the advantage or disadvantages of the flakes vs dissolved chitosan? At least that could act as a control.

3. The discussion looks like conclusion and do not provide good discussion - only one paragraph. All in all, I could not find/understand what the authors want to achieve in the research.

Author Response

The authors thank the reviewer for carefully reading our manuscript and for the constructive criticism provided.

Our answers to the reviewer’s queries follow.

  1. The method is not clear at all. The PCl solution and the CS/PEO solution were separately prepared and separately electrospun. Why then mention the 5ml/1-ml syringe? Is it a layered system? or a biaxial system etc.

The experimental method was described as is typical for an investigation involving the production of fibrous scaffolds using electrospinning. To clarify the arrangement of the setup we added the following sentence to the experimental methods (line 128): “The PCL solution was electrospun horizontally and the PEO/CS solution was electrospun vertically.”

  1. The rationale and motivation of the study is not clear. Why are the authors using flakes... what are the advantage or disadvantages of the flakes vs dissolved chitosan? At least that could act as a control.

In our experiment the control was the PCL scaffold without CS flakes. Using dissolved chitosan (a PCL + CS solution) is possible and we studied it previously [1]. To clarify the rationale and motivation of the study, we added the following sentence (line 94): “In this study, we aimed at evaluating if electrospinning a transporter solution of a sacrificial polymer can be used to achieve a uniform and controlled incorporation of a polymer in granular form within an electrospun scaffold”

  1. The discussion looks like conclusion and do not provide good discussion - only one paragraph. All in all, I could not find/understand what the authors want to achieve in the research.

We followed the MDPI template for the journal Polymers. The conclusions section “is not mandatory but can be added to the manuscript if the discussion is unusually long or complex.” Since our discussion section is relatively short as we discussed the results along with their presentation, we did not include a “Conclusion” section.

In our opinion, the goals and results attained are adequately described in the revised manuscript.

Round 2

Reviewer 3 Report

The authors addressed most of the Reviewers' comments, and it could be published.

Author Response

We are happy that we addressed most of the Reviewers' comments and that the reviewer consideres our manuscript as fit for publication

Reviewer 4 Report

The authors have tried to provide rebuttals for the queries raised by the reviewer. I am not really satisfied with the answers. Specifically, comment 3. The authors has not clearly delineated the purpose of the composite scaffold using electrospinning. The authors have stated that "we aimed at evaluating if electrospinning a transporter solution of a sacrificial polymer can be used to achieve a uniform and controlled incorporation of a polymer in granular form within an electrospun scaffold" - however it is not clear how the PEO was used or employed as sacrificial polymer while the study do not actually sacrifice the PEO later in the manuscript/in vitro work.

The horizontal-vertical electrospinning is also not clear - is the electrospinning done simultaneously? If yes, then it may not be applicable in a rotating system. If it was done one after the other, then was it a layered system? The authors may chose to provide a schematic here.

Author Response

Please see document attached

Round 3

Reviewer 4 Report

No further comments.